# GraphMix: Regularized Training of Graph Neural Networks for Semi-Supervised Learning

## Abstract

We present *GraphMix*, a regularized training scheme for Graph Neural Network based semi-supervised object classification, leveraging the recent advances in the regularization of classical deep neural networks. Specifically, we propose a unified approach in which we train a fully-connected network jointly with the graph neural network via parameter sharing, interpolation-based regularization and self-predicted-targets. Our proposed method is architecture agnostic in the sense that it can be applied to any variant of graph neural networks which applies a parametric transformation to the features of the graph nodes. Despite its simplicity, with GraphMix we can consistently improve results and achieve or closely match state-of-the-art performance using even simpler architectures such as Graph Convolutional Networks, across three established graph benchmarks: the *Cora*, *Citeseer* and *Pubmed* citation network datasets, as well as three newly proposed datasets : *Cora-Full*, *Co-author-CS* and *Co-author-Physics*.

## 1 Introduction

Due to the presence of graph structured data across a wide variety of domains, such as biological networks, social networks and telecommunication networks, there have been several attempts to design neural networks that can process arbitrarily structured graphs. Early work includes (Gori et al.; Scarselli et al., 2009) which propose a neural network that can directly process most type of graphs e.g., acyclic, cyclic, directed, and undirected graphs. More recent approaches include (Bruna et al., 2013; Henaff et al., 2015; Defferrard et al., 2016; Kipf & Welling, 2016; Gilmer et al., 2017; Hamilton et al., 2017; Veličković et al., 2018; 2019; Qu et al., 2019; Gao & Ji, 2019; Ma et al., 2019), among others. Many of these approaches are designed for addressing the important problem of Semi-supervised learning over graph structured data (Zhou et al., 2018). Much of this research effort has been dedicated to developing novel architectures.

Unlike many existing works which try to come up with the new architectures, we re-synthesize ideas from the recent advances in the regularization of the classical neural network, and propose an architecture-agnostic framework for regularized training of graph neural network based semi-supervised object classification. Recently, Data-Augmentation based regularization has been shown to be very effective in other types of neural networks but how to apply these techniques in graph neural networks is still under-explored. Our proposed method GraphMix [1] is a unified framework that utilizes interpolation based data augmentation (Zhang et al., 2018; Verma et al., 2019a) and self-target-prediction based data-augmentation techniques (Laine & Aila, 2016; Tarvainen & Valpola, 2017; Verma et al., 2019b; Berthelot et al., 2019). We show that with our proposed unified framework, we can achieve state-of-the-art performance even when using simpler graph neural network architectures such as Graph Convolutional Networks (Kipf & Welling, 2017) and without incurring any significant additional computation cost.

---

[1] code available at https://github.com/anon777000/GraphMix

## 2 PROBLEM DEFINITION AND PRELIMINARIES

### 2.1 PROBLEM SETUP

We are interested in the problem of semi-supervised object classification using graph structured data. We can formally define such graph structured data as $\mathcal{G} = (\mathcal{V}, A)$, where $\mathcal{V}$ represents the set of nodes $\{v_1, \ldots, v_n\}$, and $A$ is the adjacency matrix representing the edges between the nodes of $\mathcal{V}$.

Each node $v_i$ in the graph has a corresponding $d$-dimensional feature vector $\mathbf{x}_i \in \mathbb{R}^d$. The feature vectors of all the nodes $\mathbf{X} = [\mathbf{x}_1, \ldots, \mathbf{x}_n]^\top$ are stacked together to form the entire feature matrix $\mathbf{X} \in \mathbb{R}^{n \times d}$. Each node belongs to one out of $C$ classes and can be labeled with a $C$-dimensional one-hot vector $\mathbf{y}_i \in \{0, 1\}^C$. Given the labels of $\mathbf{Y}_L$ for few of the labeled nodes $\mathcal{V}_L \subset \mathcal{V}$, the task is to predict the labels $\mathbf{Y}_U$ of the remaining nodes $\mathcal{V}_U = \mathcal{V} \setminus \mathcal{V}_L$.

### 2.2 GRAPH NEURAL NETWORKS

Graph Neural Networks (GNN) learn the $l_{th}$ layer representations of a sample $i$ by leveraging the representations of the samples $NB(i)$ in the neighbourhood of $i$. This is done by using an aggregation function that takes as an input the representations of all the samples and the graph structure and outputs the aggregated representation. The aggregation function can be defined using the Graph Convolution layer (Kipf & Welling, 2017), Graph Attention Layer (Veličković et al., 2018), or any general message passing layer (Gilmer et al., 2017). Formally, let $\mathbf{H}_l \in \mathbb{R}^{n \times k}$ be a matrix containing the $k$-dimensional representation of $n$ nodes in the $l_{th}$ layer, then:

$$\mathbf{H}_{l+1} = a(\mathbf{H}_l \mathbf{W}, A) \tag{1}$$

where $\mathbf{W} \in \mathbb{R}^{k \times k'}$ is a linear transformation matrix, $k'$ is the dimension of $(l+1)_{th}$ layer and $a$ is the aggregation function that utilizes the graph adjacency matrix $A$.

### 2.3 INTERPOLATION BASED REGULARIZATION TECHNIQUES

Recently, interpolation-based techniques have been proposed for regularizing neural networks. We briefly describe some of these techniques here. Mixup (Zhang et al., 2018) trains a neural network on the convex combination of input and targets, whereas Manifold Mixup (Verma et al., 2019a) trains a neural network on the convex combination of the hidden states of a randomly chosen hidden layer and the targets. While Mixup regularizes a neural network by enforcing that the model output should change linearly in between the examples in the input space, Manifold Mixup regularizes the neural network by learning better (more discriminative) hidden states.

Formally, suppose $g : \mathbf{x} \to \mathbf{h}$ is a function that maps input sample to hidden states, $f : \mathbf{h} \to \hat{\mathbf{y}}$ is a function that maps hidden states to predicted output, $\lambda$ is a random variable drawn from $\text{Beta}(\alpha, \alpha)$ distribution, $\text{Mix}_\lambda(\mathbf{a}, \mathbf{b}) = \lambda * \mathbf{a} + (1 - \lambda) * \mathbf{b}$ is an interpolation function, $\mathcal{D}$ is the data distribution, $(\mathbf{x}, \mathbf{y})$ and $(\mathbf{x}', \mathbf{y}')$ is a pair of examples sampled from distribution $\mathcal{D}$ and $\ell$ be a loss function such as cross-entropy loss, then the Manifold Mixup Loss is defined as:

$$\mathcal{L} = \underset{(\mathbf{x},\mathbf{y}) \sim \mathcal{D}}{\mathbb{E}} \; \underset{(\mathbf{x}',\mathbf{y}') \sim \mathcal{D}}{\mathbb{E}} \; \underset{\lambda \sim \text{Beta}(\alpha,\alpha)}{\mathbb{E}} \; \ell(f(\text{Mix}_\lambda(g(\mathbf{x}), g(\mathbf{x}'))), \text{Mix}_\lambda(\mathbf{y}, \mathbf{y}')). \tag{2}$$

We use above Manifold Mixup loss for training an auxiliary Fully-connected-network as described in Section 3 and Line 6 and 12 of Algorithm 1.

## 3 GRAPHMIX

### 3.1 MOTIVATION

Data Augmentation is arguably the simplest and most efficient technique for regularizing a neural network. In some domains, such as computer vision, speech and text, there exist efficient data augmentation techniques, for example, random cropping, translation or Cutout (Devries & Taylor,

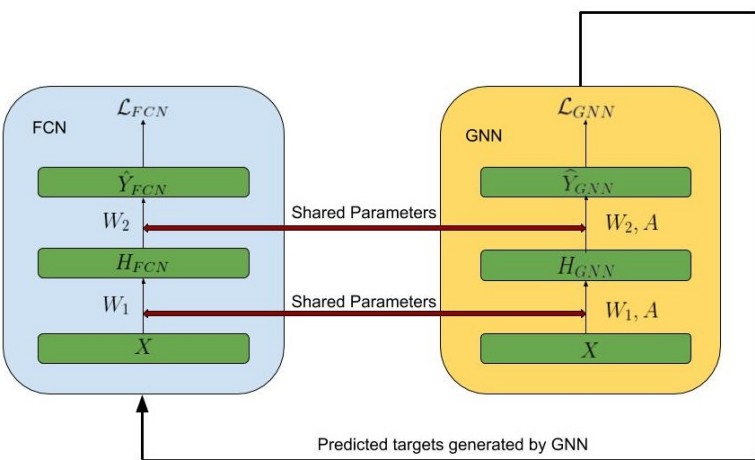

Figure 1: The procedure for training with GraphMix . The Fully-Connected Network (FCN) and the Graph Neural Network (GNN) share linear transformation matrix ($W$) applied on the node features. The FCN is trained using Manifold Mixup by interpolating the hidden states $H_{FCN}$ and the corresponding labels $Y$. This leads to better features which are transferred to the GNN via parameter sharing. The predicted targets generated by the GNN for unlabeled data are used to augment the input data for the FCN. The FCN and the GNN losses are minimized jointly by alternate minimization.

2017) for computer vision, Ko et al. (2015) and Park et al. (2019) for speech and Xie et al. (2017) for text domain. However, data augmentation for the graph-structured data remains under-explored. There exists some recent work along these lines but the prohibitive additional computation cost (see Section 5.3) introduced by these methods make them impractical for real-world large graph datasets. Based on these limitations, our main objective is to propose an efficient data augmentation technique for graph datasets.

Recent work based on interpolation-based data augmentation (Zhang et al., 2018; Verma et al., 2019a) has seen sizable improvements in regularization performance across a number of tasks. However, these techniques are not directly applicable to graphs for an important reason: Although we can create additional nodes by interpolating the features and corresponding labels, it remains unclear how these new nodes must be connected to the original nodes via synthetic edges such that the structure of the whole graph is preserved. To alleviate this issue, we propose to train an auxiliary Fully-connected-net (FCN) using Manifold Mixup as discussed in Section 3.2. Note that the FCN only uses the node features ( not the graph structure), thus the Manifold mixup loss in Eq. 2 can be directly used for training the FCN. Furthermore, drawing inspiration from the success of self-supervised semi-supervised learning algorithms (self-predicted-targets based algorithms which can be also interpreted as a form of data-augmentation techniques) (Verma et al., 2019b; Berthelot et al., 2019), we explore self-supervision in the training of GNNs. We note that self-supervision has already been explored for unsupervised representation learning from graph structured data (Veličković et al., 2019), but not for semi-supervised object classification over graph structured data. Based on these challenges and motivations we present our proposed approach GraphMix for training Graph Neural Networks in the following Section.

## 3.2 METHOD

GraphMix augments the vanilla GNN with a Fully Connected Network (FCN). The FCN loss is computed using the *Manifold Mixup* as discussed in Section 2.3 and the GNN loss is computed in the standard way. The *Manifold Mixup* training of FCN facilitates learning more discriminative node representations. How to transfer these discriminative node representations to the GNN? We apply parameter sharing between FCN and GNN to facilitate this. Using these more discriminative representations of the nodes, as well as the graph structure, GNN loss is computed in the usual way.

---

**Algorithm 1** GraphMix : A procedure for improved training of Graph Neural Networks (GNN)

---

1: **Input:** A GCN: $g(X, A, \theta)$, a FCN: $f(X, \theta, \lambda)$ which shares parameters with the GCN. Beta distribution parameter $\alpha$ for *Manifold Mixup* . Number of random perturbations $K$, Sharpening temperature $T$. Consistency parameter $\gamma$. Number of epochs $N$. $\gamma(t)$: rampup function for increasing the importance of consistency regularization. $(X_L, Y_L)$ represents labeled samples and $X_U$ represents unlabeled samples.

2: **for** $t = 1$ **to** $N$ **do**

3:     i = random(0,1)   // *generate randomly 0 or 1*

4:     **if** i=0 **then**

5:        $\lambda \sim \text{Beta}(\alpha, \alpha)$   // *Sample a mixing coefficient from Beta distribution*

6:        $\mathcal{L}_{sup} = \mathcal{L}\big(f(X_L, \theta, \lambda), Y_L\big)$   // *supervised loss from FCN using the Manifold Mixup*

7:        **for** $k = 1$ **to** $K$ **do**

8:           $\hat{X}_{U,k} = RandomPerturbations(X_U)$   // *Apply $k^{th}$ round of random perturbation to $X_U$*

9:        **end for**

10:       $\bar{Y}_U = \frac{1}{K} \sum_k g(Y \mid \hat{X}_{U,k}; \theta, A)$   // *Compute average predictions across K perturbations of $X_U$ using the GCN*

11:       $Y_U = \text{Sharpen}(\bar{Y}_U, T)$   // *Apply temperature sharpening to the average prediction*

12:       $\mathcal{L}_{usup} = \mathcal{L}\big(f(X_U, \theta, \lambda), Y_U\big)$   // *unsupervised loss from FCN using the Manifold Mixup*

13:       $\mathcal{L} = \mathcal{L}_{sup} + \gamma(t) * \mathcal{L}_{usup}$   // *Total loss is the weighted sum of supervised and unsupervised FCN loss*

14:     **else**

15:        $\mathcal{L} = \mathcal{L}\big(g(X_L, \theta, A), Y_L\big)$   // *Loss using the vanilla GCN*

16:     **end if**

17: **end for**

18: **return** $\mathcal{L}$

---

Both the FCN loss and GNN loss are optimized in an alternating fashion during training. Furthermore, the predicted targets from the GNN are used to augment the training set of the FCN. In this way, both FCN and GNN facilitate each other's learning process. At inference time, the predictions are made using only GNN. The diagrammatic representation of GraphMix is presented in Figure 1 and the full algorithm is presented in Algorithm 1.

The GraphMix framework can be applied to any underlying GNN as long as the underlying GNN applies parametric transformations to the node features. In our experiments, we show the improvements over GCN (Kipf & Welling, 2016), GAT (Veličković et al., 2018) and Graph U-Net using GraphMix . Furthermore, GraphMix(GCN) framework does not add any *significant computation cost* over the underlying GNN, because the underlying GNN is trained in the standard way and the FCN training requires trivial additional computation cost for computing the predicted-targets (Section 3.2.1 and 3.2.1) and the interpolation function ( $\text{Mix}_\lambda(\mathbf{a}, \mathbf{b})$ in Section 2.3). There are no additional memory requirements for GraphMix(GCN) , since FCN and GNN share the parameters.

Some implementation considerations. For *Manifold Mixup* training of FCN, we apply *mixup* only in the hidden layer. Note that in Verma et al. (2019a), the authors recommended applying mixing in a randomly chosen layer (which also includes the input layer) at each training update. However, we observed under-fitting when applying *mixup* randomly at the input layer or hidden layer. Applying *mixup* only in the input layer also resulted in underfitting and did not improve test accuracy.

The performance of self-supervision based algorithms such as GraphMix is greatly affected by the accuracy of the predicted targets. To improve the accuracy of the predicted targets, we applied the average of the model prediction on $K$ random perturbations of an input sample as discussed in Section 3.2.1 and sharpening as described in Section 3.2.2. Further, we draw similarities and difference of GraphMix w.r.t. Co-training framework in the Section 3.2.3.

### 3.2.1 ACCURATE TARGET PREDICTION FOR UNLABELED DATA

Recent state-of-the-art semi-supervised learning methods use a *teacher* model to accurately predict targets for the unlabeled data. These predicted targets on the unlabeled data are used as "true labels" for further training of the model. The teacher model can be realized as a temporal ensemble of the *student* model (the model being trained) (Laine & Aila, 2016) or by using an Exponential Moving Average (EMA) of the parameters of the student model (Tarvainen & Valpola, 2017). Another recently proposed method for accurate target predictions for unlabeled data is to use the average of the predicted targets across $K$ random augmentations of the input sample (Berthelot et al., 2019).

Along these lines, in this work, we compute the predicted target as the average of predictions on $K$ drop-out versions of the input sample. We also used the EMA of the student model but it did not improve test accuracy across all the datasets (see Section 4.4 for details).

### 3.2.2 ENTROPY MINIMIZATION

Many recent semi-supervised learning algorithms (Laine & Aila, 2016; Miyato et al., 2018; Tarvainen & Valpola, 2017; Verma et al., 2019b) are based on the cluster assumption (Chapelle et al., 2010), which posits that the class boundary should pass through the low-density regions of the marginal data distribution. One way to enforce this assumption is to explicitly minimize the entropy of the model's predictions $p(y|x, \theta)$ on unlabeled data by adding an extra loss term to the original loss term (Grandvalet & Bengio, 2005). The entropy minimization can be also achieved implicitly by modifying the model's prediction on the unlabeled data such that the prediction has low entropy and using these low-entropy predictions as targets for the further training of the model. Examples include "Pseudolabels" (Lee, 2013) and "Sharpening" (Berthelot et al., 2019). Pseudolabeling constructs hard(1-hot) labels for the unlabeled samples which have "high-confidence predictions". Since many of the unlabeled samples may have "low-confidence predictions", they can not be used in the Pseudolabeling technique. On the other hand, Sharpening does not require "high-confidence predictions", and thus it can be used for all the unlabelled samples. Hence in this work, we use Sharpening for entropy minimization. The Sharpening function over the model prediction $p(y|x, \theta)$ can be formally defined as follows (Berthelot et al., 2019), where $T$ is the temperature hyperparameter and $C$ is the number of classes:

$$\text{Sharpen}(p_i, T) := p_i^{\frac{1}{T}} \bigg/ \sum_{j=1}^{C} p_j^{\frac{1}{T}} \tag{3}$$

### 3.2.3 CONNECTION TO CO-TRAINING

The GraphMix approach can be seen as a special instance of the Co-training framework (Blum & Mitchell, 1998). Co-training assumes that the description of an example can be partitioned into two *distinct* views and either of these views would be sufficient for learning if we had enough labeled data. In this framework, two learning algorithms are trained separately on each view and then the prediction of each learning algorithm on the unlabeled data is used to enlarge the training set of the other. Our method has some important differences and similarities to the Co-training framework. Similar to Co-training, we train two neural networks and the predictions from the GNN are used to enlarge the training set of FCN. The important difference is that instead of using the predictions from the FCN to enlarge the training set for the GNN, we employ parameter sharing for passing the learned information from FCN to GNN. In our experiments, directly using the predictions of the FCN for GNN training resulted in reduced accuracy. This is due to the fact that the number of labeled samples for training the FCN is sufficiently low and hence the FCN does not make accurate enough predictions. Another important difference is that unlike the co-training framework, FCN and GNN do not use completely distinct views of the data: the FCN uses feature vectors $\mathbf{X}$ and the GNN uses the feature vector and adjacency matrix $(\mathbf{X}, A)$.

## 4 EXPERIMENTS

We present results for the GraphMix algorithm using standard benchmark datasets and the standard architecture in Section 4.2 and 4.3. We also conduct an ablation study on GraphMix in Section 4.4 to understand the contribution of various components to its performance. Refer to Appendix A.5 for implementation and hyperparameter details.

### 4.1 DATASETS

We use three standard benchmark citation network datasets for semi-supervised node classification, namely Cora, Citeseer and Pubmed. In all these datasets, nodes correspond to documents and edges correspond to citations. Node features correspond to the bag-of-words representation of the document. Each node belongs to one of $C$ classes. During training, the algorithm has access to the feature vectors and edge connectivity of all the nodes but has access to the class labels of only a few of the nodes.

For semi-supervised link classification, we use two datasets Bitcoin Alpha and Bitcoin OTC from (Kumar et al., 2016; 2018). The nodes in these datasets correspond to the bitcoin users and the edge weights between them correspond to the degree of trust between the users. Following (Qu et al., 2019), we treat edges with weights greater than 3 as positive instances, and edges with weights less than -3 are treated as negative ones. Given a few labeled edges, the task is to predict the labels of the remaining edges. The statistics of these datasets as well as the number of training/validation/test nodes is presented in Appendix A.1.

## 4.2 SEMI-SUPERVISED NODE CLASSIFICATION

For baselines, we choose GCN (Kipf & Welling, 2017), and the recent state-of-the-art methods GAT (Veličković et al., 2018), GMNN (Qu et al., 2019) and Graph U-Net (Gao & Ji, 2019). We additionally use two self-training based baselines: in the first one, we trained a GCN with self-generated predicted targets, and in the second one, we trained a FCN with self-generated predicted targets, named "GCN (with predicted-targets)" and "FCN (with predicted-targets)" respectively in Table 1. For generating the predicted-targets in above two baselines, we followed the procedure of Section 3.2.1 and Section 3.2.2. GraphMix(GCN) , GraphMix(GAT) and GraphMix(Graph U-Net) refer to the methods where underlying GNNs are GCN, GAT and Graph U-Net respectively. Refer to Appendix Section A.5 for implementation and hyperparameter details.

Shchur et al. (2018) has demonstrated that the performance of the current state-of-the-art Graph Neural Networks on the standard train/validation/test split of the popular benchmark datasets (such as Cora, Citeseer, Pubmed, etc) is significantly different from their performance on the random splits. For fair evaluation, they recommend using multiple random partitions of the datasets. Along these lines, we created 10 random splits of the Cora, Citeseer and Pubmed with the same train/ validation/test number of samples as in the standard split. We also provide the results for the standard train/validation/test split in Table 5 in Appendix A.2.

The results are presented in Table 1. We observe that GraphMix always improves the accuracy of the underlying GNNs such as GCN, GAT and Graph U-Net across all the dataset, with GraphMix(GCN) achieving the best results.

We further present results with fewer labeled samples and results on larger datasets (Cora-Full, Co-author-CS and Co-author-Physics) Section A.3 and Section A.4 respectively.

Table 1: Results of node classification (% test accuracy) using 10 random Train/Validation/Test split of datasets. [*] means the results are taken from the corresponding papers. We conduct 100 trials and report mean and standard deviation over the trials (refer to Table 5 in the Appendix for comparison with other methods on standard Train/Validation/Test split).

| Algorithm | Cora | Citeseer | Pubmed |
|---|---|---|---|
| GCN | 77.84±1.45 | 72.56±2.46 | 78.74±0.99 |
| GAT | 77.74±1.86 | 70.41±1.81 | 78.48±0.96 |
| Graph U-Net | 77.59±1.60 | 67.55±0.69 | 76.79±2.45 |
| GCN(with predicted-targets) | 80.41±1.78 | 73.62±2.11 | 79.81±2.85 |
| FCN(with predicted-targets) | 75.19±3.53 | 70.49±1.91 | 73.40±2.48 |
| GraphMix (GCN) | **82.07±1.17** | **76.45±1.57** | **80.72±1.08** |
| GraphMix (GAT) | 80.63±1.31 | 74.08±1.26 | 80.14±1.51 |
| GraphMix (Graph-U-Net) | 80.18±1.62 | 72.85±1.71 | 78.47±0.64 |

## 4.3 SEMI-SUPERVISED LINK CLASSIFICATION

In the Semi-supervised Link Classification problem, the task is to predict the labels of the remaining links, given a graph and labels of a few links. Following (Taskar et al., 2004), we can formulate the link classification problem as a node classification problem. Specifically, given an original graph $G$, we construct a dual Graph $G'$. The node set $V'$ of the dual graph corresponds to the link set $E'$ of the original graph. The nodes in the dual graph $G'$ are connected if their corresponding links in the graph $G$ share a node. The attributes of a node in the dual graph are defined as the index of the

nodes of the corresponding link in the original graph. Using this formulation, we present results on link classification on Bit OTC and Bit Alpha benchmark datasets in the Table 2. As the numbers of the positive and negative edges are strongly imbalanced, we report the F1 score. Our results show that GraphMix(GCN) improves the performance over the baseline GCN method for both the datasets. Furthermore, the results of GraphMix(GCN) are comparable with the recently proposed state-of-the-art method GMNN (Qu et al., 2019).

Table 2: Results on Link Classification (%F1 score). [*] means the results are taken from the corresponding papers

| Algorithm | Bit OTC | Bit Alpha |
|---|---|---|
| DeepWalk (Perozzi et al., 2014) | 63.20 | 62.71 |
| GMNN*(Qu et al., 2019) | **66.93** | **65.86** |
| GCN | 65.72±0.38 | 64.00±0.19 |
| GCN(with predicted-targets) | 65.15±0.29 | 64.56±0.21 |
| FCN(with predicted-targets) | 60.13±0.40 | 59.74±0.32 |
| GraphMix (GCN) | 66.35±0.41 | 65.34±0.19 |

## 4.4 ABLATION STUDY

Since GraphMix consists of various components, some of which are common with the existing literature of semi-supervised learning, we set out to study the effect of various components by systematically removing or adding a component from GraphMix . We measure the effect of the following:

- Removing the Manifold Mixup and predicted targets from the FCN training.

- Removing the predicted targets from the FCN training.

- Removing the Manifold Mixup from the FCN training.

- Removing the Sharpening of the predicted targets.

- Removing the Average of predictions for $K$ random perturbations of the input sample

- Using the EMA (Tarvainen & Valpola, 2017) of GNN for target prediction.

The ablation results for semi-supervised node classification are presented in Table 3. We did not do any hyperparameter tuning for the ablation study and used the best performing hyperparameters found for the results presented in Table 1. We observe that all the components of GraphMix contribute to its performance, with Manifold Mixup training of FCN contributing possibly the most. Furthermore, we observe that using the EMA model (which is an emsemble model) (Tarvainen & Valpola, 2017) for computing the predicted- targets could improve the performance of GraphMix for all the datasets.

Table 3: Ablation study results using 10 labeled samples per class (% test accuracy). We report mean and standard deviation over ten trials.

| Ablation | Cora | Citeseer | Pubmed |
|---|---|---|---|
| GraphMix | 79.30±1.36 | 70.78±1.41 | 77.13±3.60 |
| -without Manifold Mixup, without predicted targets | 68.78±3.54 | 61.01±1.24 | 72.56±1.08 |
| -without predicted-targets | 72.85±3.79 | 64.40±2.20 | 74.74±1.69 |
| -without Manifold Mixup | 69.08±5.03 | 62.66±1.80 | 74.11±0.94 |
| -no Sharpening | 73.25±3.41 | 64.65±2.21 | 74.97±1.47 |
| -no Averaging of predictions | 74.17±1.99 | 65.52±1.78 | 75.59±2.63 |
| -with EMA | 79.84±2.28 | 71.21±1.32 | 77.46±3.13 |

### 4.5 Visualization of the Learned Features

In this section, we present the analysis of the features learned by GraphMix for Cora dataset. Specifically, we present the 2D visualization of the hidden states using the t-SNE (van der Maaten & Hinton, 2008) in Figure 2a, 2b and 2c . We observe that GraphMix learns hidden states which are better separated and condensed than GCN and GCN(predicted-targets). We further evaluate the Soft-rank (refer to Appendix A.7) of the class-specific hidden states to demonstrate that GraphMix(GCN) makes the class-specific hidden states more concentrated than GCN and GCN(predicted-targets), as shown in 2d. Refer to Appendix A.8 for 2D representation of other datasets.

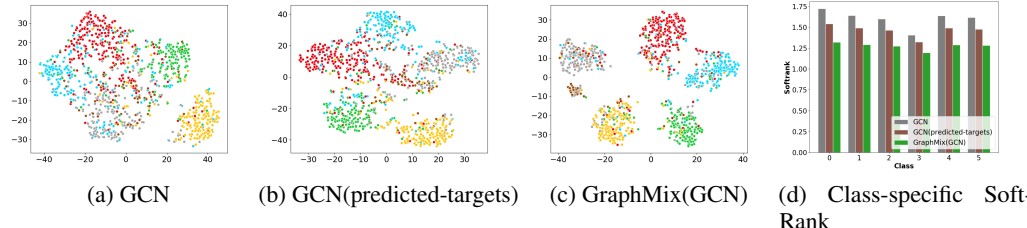

| (a) GCN | (b) GCN(predicted-targets) | (c) GraphMix(GCN) | (d) Class-specific Soft-Rank |

Figure 2: 2D representation of the hidden states of Citeseer dataset using (a) GCN, (b) GCN(predicted-targets), (c) GraphMix, and (d) Soft-Rank of Class-specific hidden states (lower Soft-Rank reflects more concentrated class-specific hidden states)

## 5 Related Work

### 5.1 Semi-supervised Learning over Graph Data

There exists a long line of work for Semi-supervised learning over Graph Data. Earlier work included using *Graph Laplacian Regularizer* for enforcing local smoothness over the predicted targets for the nodes (Zhu & Ghahramani, 2002; Zhu et al., 2003; Belkin et al., 2006). Another line of work learns node embedding in an unsupervised way (Perozzi et al., 2014) which can then be used as an input to any classifier, or learns the node embedding and target prediction jointly (Yang et al., 2016). Many of the recent Graph Neural Network based approaches (refer to Zhou et al. (2018) for a review of these methods) are inspired by the success of Convolutional Neural Networks in image and text domains, defines the convolutional operators using the neighbourhood information of the nodes (Kipf & Welling, 2017; Veličković et al., 2018; Defferrard et al., 2016). These convolution operator based method exhibit state-of-the-results for semi-supervised learning over graph data, hence much of the recent attention is dedicated to proposing architectural changes to these methods (Qu et al., 2019; Gao & Ji, 2019; Ma et al., 2019). Unlike these methods, we propose a regularization technique that can be applied to any of these Graph Neural Networks which uses a parameterized transformation on the node features.

### 5.2 Data Augmentation

It is well known that the generalization of a learning algorithm can be improved by enlarging the training data size. Because labeling more samples is labour-intensive and costly, Data-augmentation has become *de facto* technique for enlarging the training data size, especially in the computer vision applications such as image classification. Some of the notable Data Augmentation techniques include Cutout (Devries & Taylor, 2017) and DropBlock (Ghiasi et al., 2018). In Cutout, a contiguous part of the input is zeroed out. DropBlock further extends Cutout to the hidden states. In another line of research, such as Mixup and BC-learning (Zhang et al., 2018; Tokozume et al., 2017), additional training samples are generated by interpolating the samples and their corresponding targets. Manifold Mixup (Verma et al., 2019a) proposes to augment the data in the hidden states and shows that it learns more *discriminative* features for supervised learning. Furthermore, ICT (Verma et al., 2019b) and MixMatch (Berthelot et al., 2019) extend the Mixup technique to semi-supervised learning, by computing the predicted targets for the unlabeled data and applying the Mixup on the unlabeled data

and their corresponding predicted targets. Even further, for unsupervised learning, ACAI (Berthelot* et al., 2019) and AMR (Beckham et al., 2019) explore the interpolation techniques for autoencoders. ACAI interpolates the hidden states of an autoencoder and uses a critic network to constrain the reconstruction of these interpolated states to be realistic. AMR explores different ways of combining the hidden states of an autoencoder other than the convex combinations of the hidden states. Unlike, all of these techniques which have been proposed for the fixed topology datasets, in this work, we propose interpolation based data-augmentation techniques for graph structured data.

### 5.3 REGULARIZING GRAPH NEURAL NETWORKS

Regularizing Graph Neural Networks has drawn some attention recently. GraphSGAN (Ding et al., 2018) first uses an embedding method such as DeepWalk (Perozzi et al., 2014) and then trains generator-classifier networks in the adversarial learning setting to generate fake samples in the low-density region between sub-graphs. BVAT (Deng et al., 2019) and Feng et al. (2019) generate adversarial perturbations to the features of the graph nodes while taking graph structure into account. While these methods improve generalization in graph-structured data, they introduce significant additional computation cost: GraphScan requires computing node embedding as a preprocessing step, BVAT and Feng et al. (2019) require additional gradient computation for computing adversarial perturbations. Unlike these methods, GraphMix does not introduce any significant additional computation since it is based on interpolation-based techniques and self-generated predicted targets.

## 6 DISCUSSION

We presented GraphMix , a simple and efficient regularized training scheme for graph neural networks. GraphMix is a general scheme that can be applied to any graph neural network that uses a parameterized transformation on the feature vector of the graph nodes. Through extensive experiments, we demonstrated state-of-the-art performances or close to state-of-the-art performance using this simple regularization technique on various benchmark datasets, more importantly, GraphMix improves test accuracy over vanilla GNN across all the datasets, even without doing any extensive hyperparameter search. Further, we conduct a systematic ablation study to understand the effect of different components in the performance of GraphMix . This suggests that in parallel to designing new architectures, exploring better regularization for graph structured data is a promising avenue for research.

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

# A  APPENDIX

## A.1  DATASETS

The statistics of these datasets as well as the number of training/validation/test nodes is presented in Table 4.

Table 4: Dataset statistics.

| Dataset | # Nodes | # Edges | # Features | # Classes | # Training | # Validation | # Test |
|---------|---------|---------|-----------|----------|-----------|-------------|--------|
| Cora | 2,708 | 5,429 | 1,433 | 7 | 140 | 500 | 1,000 |
| Citeseer | 3,327 | 4,732 | 3,703 | 6 | 120 | 500 | 1,000 |
| Pubmed | 19,717 | 44,338 | 500 | 3 | 60 | 500 | 1,000 |
| Bitcoin Alpha | 3,783 | 24,186 | 3,783 | 2 | 100 | 500 | 3,221 |
| Bitcoin OTC | 5,881 | 35,592 | 5,881 | 2 | 100 | 500 | 5,947 |

## A.2  COMPARISON WITH STATE-OF-THE-ART METHODS

We present the comparion of GraphMix with the recent state-of-the-art methods as well as earlier methods using the standard Train/Validation/Test split in Table 5.

## A.3  RESULTS WITH FEWER LABELED SAMPLES

We further evaluate the effectiveness of GraphMix in the learning regimes where fewer labeled samples exist. For each class, we randomly sampled $K \in \{5, 10\}$ samples for training and the same number of samples for the validation. We used all the remaining labeled samples as the test set. We repeated this process for 10 times. The results in Table 6 show that GraphMix achieves even better improvements when the labeled samples are fewer ( Refer to Table 1 for results with 20 training samples per class).

## A.4  RESULTS ON LARGER DATASETS

In this section, we provide results on three recently proposed datasets which are relatively larger than standard benchmark datasets (Cora/Citeseer/Pubmed). Specifically, we use Cora-Full dataset proposed in Bojchevski & Günnemann (2018) and Coauthor-CS and Coauthor-Physics datasets proposed in Shchur et al. (2018). We took processed versions of these dataset available here [2]. We did 10 random splits of the the data into train/validation/test split. For the classes which had more than 100 samples, we choose 20 samples per class for training, 30 samples per class for validation and the remaining samples as test data. For the classes which had less than 100 samples, we chose 20% samples, per class for training, 30% samples for validation and the remaining for testing. For each split we run experiments using 100 random seeds. The statistics of these datasets in presented in Table 8 and the results are presented in Table 7. We observe that GraphMix(GCN) improves the results over GCN for all the three datasets. We note that we did minimal hyperparameter search for GraphMix(GCN) as mentioned in Section A.8.1, and doing more rigorous hyperparameter search can further improve the performance of GraphMix .

---

[2]https://github.com/shchur/gnn-benchmark

Table 5: Comparison of GraphMix with other methods (% test accuracy ), for Cora, Citeseer and Pubmed.

| Method | Cora | Citeseer | Pubmed |
|---|---|---|---|
| Results reported from the literature | | | |
| MLP | 55.1% | 46.5% | 71.4% |
| ManiReg (Belkin et al., 2006) | 59.5% | 60.1% | 70.7% |
| SemiEmb (Weston et al., 2012) | 59.0% | 59.6% | 71.7% |
| LP (Zhu et al., 2003) | 68.0% | 45.3% | 63.0% |
| DeepWalk (Perozzi et al., 2014) | 67.2% | 43.2% | 65.3% |
| ICA (Lu & Getoor, 2003) | 75.1% | 69.1% | 73.9% |
| Planetoid (Yang et al., 2016) | 75.7% | 64.7% | 77.2% |
| Chebyshev (Defferrard et al., 2016) | 81.2% | 69.8% | 74.4% |
| GCN (Kipf & Welling, 2017) | 81.5% | 70.3% | 79.0% |
| MoNet (Monti et al., 2016) | $81.7 \pm 0.5\%$ | — | $78.8 \pm 0.3\%$ |
| GAT (Veličković et al., 2018) | $83.0 \pm 0.7\%$ | $72.5 \pm 0.7\%$ | $79.0 \pm 0.3\%$ |
| GraphScan (Ding et al., 2018) | $83.3 \pm 1.3$ | $73.1 \pm 1.8$ | — |
| GMNN (Qu et al., 2019) | 83.7% | 73.1% | **81.8%** |
| DisenGCN (Ma et al., 2019) | 83.7% | 73.4% | 80.5% |
| Graph U-Net (Gao & Ji, 2019) | **84.4%** | 73.2% | 79.6% |
| BVAT (Deng et al., 2019) | $83.6 \pm 0.5$ | **$74.0 \pm 0.6$** | $79.9 \pm 0.4$ |
| Our Experiments | | | |
| GCN | $81.30 \pm 0.66$ | $70.61 \pm 0.22$ | $79.86 \pm 0.34$ |
| GAT | $82.70 \pm 0.21$ | $70.40 \pm 0.35$ | $79.05 \pm 0.64$ |
| Graph U-Net | $81.74 \pm 0.54$ | $67.69 \pm 1.10$ | $77.73 \pm 0.98$ |
| GCN (with predicted-targets) | $82.03 \pm 0.43$ | $73.38 \pm 0.35$ | **$82.42 \pm 0.36$** |
| FCN (with predicted-targets) | $80.30 \pm 0.75$ | $71.50 \pm 0.80$ | $77.40 \pm 0.37$ |
| GraphMix (GCN) | **$83.94 \pm 0.57$** | **$74.52 \pm 0.59$** | $80.98 \pm 0.55$ |
| GraphMix (GAT) | $83.32 \pm 0.18$ | $73.08 \pm 0.23$ | $81.10 \pm 0.78$ |
| GraphMix (Graph U-Net) | $82.18 \pm 0.63$ | $69.00 \pm 1.32$ | $78.76 \pm 1.09$ |

Table 6: Results using less labeled samples (% test accuracy). $K$ referes to the number of labeled samples per class.

| Algorithm | Cora | | Citeseer | | Pubmed | |
|---|---|---|---|---|---|---|
| | $K = 5$ | $K = 10$ | $K = 5$ | $K = 10$ | $K = 5$ | $K = 10$ |
| GCN | $66.39 \pm 4.26$ | $72.91 \pm 3.10$ | $55.61 \pm 5.75$ | $64.19 \pm 3.89$ | $66.06 \pm 3.85$ | $75.57 \pm 1.58$ |
| GAT | $68.17 \pm 5.54$ | $73.88 \pm 4.35$ | $55.54 \pm 1.82$ | $61.63 \pm 0.42$ | $64.24 \pm 4.79$ | $73.60 \pm 1.85$ |
| Graph U-Net | $64.42 \pm 5.44$ | $71.48 \pm 3.03$ | $49.43 \pm 5.81$ | $61.16 \pm 3.47$ | $65.05 \pm 4.69$ | $68.65 \pm 3.69$ |
| GraphMix (GCN) | **$71.99 \pm 6.46$** | **$79.30 \pm 1.36$** | **$58.55 \pm 2.26$** | **$70.78 \pm 1.41$** | **$67.66 \pm 3.90$** | **$77.13 \pm 3.60$** |
| GraphMix (GAT) | $72.01 \pm 6.68$ | $75.82 \pm 2.73$ | $57.6 \pm 0.64$ | $62.24 \pm 2.90$ | $66.61 \pm 3.69$ | $75.96 \pm 1.70$ |
| GraphMix (Graph U-Net) | $66.84 \pm 6\ 5.10$ | $73.14 \pm 3.17$ | $54.39 \pm 5.07$ | $64.36 \pm 3.48$ | $67.40 \pm 5.33$ | $70.43 \pm 3.75$ |

## A.5 IMPLEMENTATION AND HYPERPARAMETER DETAILS

We use the standard benchmark architecture as used in GCN (Kipf & Welling, 2017), GAT (Veličković et al., 2018) and GMNN (Qu et al., 2019), among others. This architecture has one hidden layer and the graph convolution is applied twice : on the input layer and on the output of the hidden layer. The FCN in GraphMix shares the parameters with the GCN.

GraphMix introduces four additional hyperparameters, namely the $\alpha$ parameter of Beta distribution used in Manifold Mixup training of the FCN, the max-consistency coefficient $\gamma_{max}$ which controls the trade-off between the supervised loss and the unsupervised loss (loss computed using the pseudolables)

Table 7: Comparison of GraphMix with other methods (% test accuracy ), for Cora-Full, Coauthor-CS, Coauthor-Physics, and NELL. $*$ refers to the results reported in Shchur et al. (2018).

| Method | Cora-Full | Coauthor-CS | Coauthor-Physics | NELL |
|--------|-----------|-------------|------------------|------|
| GCN* | **62.2±0.6** | 91.1±0.5 | 92.8±1.0 | — |
| GAT* | 51.9±1.5 | 90.5±0.6 | 92.5±0.9 | — |
| MoNet* | 59.8±0.8 | 90.8±0.6 | 92.5±0.9 | — |
| GS-Mean* | 58.6±1.6 | 91.3±2.8 | 93.0±0.8 | — |
| GCN (Kipf & Welling, 2017) | — | — | — | 66.0 |
| GCN | 60.13±0.57 | 91.27±0.56 | 92.90±0.92 | 63.65±1.17 |
| GraphMix (GCN) | 61.80±0.54 | **91.83±0.51** | **94.49±0.84** | **66.32±1.04** |

Table 8: Dataset statistics

| Datasets | Classes | Features | Nodes | Edges |
|----------|---------|----------|-------|-------|
| Cora-Full | 67 | 8710 | 18703 | 62421 |
| Coauthor-CS | 15 | 6805 | 18333 | 81894 |
| Coauthor-Physics | 5 | 8415 | 34493 | 247962 |
| NELL | 210 | 5414 | 65755 | 266144 |

of FCN, the temparature $T$ in sharpening and the number of random perturbations $K$ applied to the input data for the averaging of the predictions.

We conducted minimal hyperparameter seach over only $\alpha$ and $\gamma_{max}$ and fixed the hyperparameters $T$ and $K$ to $0.1$ and $10$ respectively. The other hyperparameters were set to the best values for underlying GNN (GCN or GAT), including the learning rate, the $L2$ decay rate, number of units in the hidden layer etc. We observed that GraphMix is not very sensitive to the values of $\alpha$ and $\gamma_{max}$ and similar values of these hyperparameters work well across all the benchmark datasets. Refer to Appendix A.5 and A.6 for the details about the hyperparameter values and the procedure used for the best hyperparameters selection.

### A.5.1 FOR RESULTS REPORTED IN SECTION 4.2

For GCN and GraphMix(GCN), we used Adam optimizer with learning rate $0.01$ and $L2$-decay 5e-4, the number of units in the hidden layer $16$ , dropout rate in the input layer and hidden layer was set to $0.5$ and $0.0$, respectively. For GAT and GraphMix(GAT), we used Adam optimizer with learning rate $0.005$ and $L2$-decay 5e-4, the number of units in the hidden layer $8$ , and the dropout rate in the input layer and hidden layer was searched from the values $\{0.2, 0.5, 0.8\}$.

For $\alpha$ and $\gamma_{max}$ of GraphMix(GCN) and GraphMix(GAT) , we searched over the values in the set $[0.0, 0.1, 1.0, 2.0]$ and $[0.1, 1.0, 10.0, 20.0]$ respectively.

For GraphMix(GCN) : $\alpha = 1.0$ works best across all the datasets. $\gamma_{max} = 1.0$ works best for Cora and Citeseer and $\gamma_{max} = 10.0$ works best for Pubmed.

For GraphMix(GAT) : $\alpha = 1.0$ works best for Cora and Citeseer and $\alpha = 0.1$ works best for Pubmed. $\gamma_{max} = 1.0$ works best for Cora and Citeseer and $\gamma_{max} = 10.0$ works best for Pubmed. Input droputrate=0.5 and hidden dropout rate=0.5 work best for Cora and Citeseer and Input dropout rate=0.2 and hidden dropout rate =0.2 work best for Pubmed.

We conducted all the experiments for 2000 epochs. The value of consistency coefficient $\gamma$ (line 13 in Algorithm 1) is increased from 0 to its maximum value $\gamma_{max}$ from epoch 500 to 1000 using the sigmoid ramp-up of Mean-Teacher (Tarvainen & Valpola, 2017).

### A.5.2 FOR RESULTS REPORTED IN SECTION A.3

For $\alpha$ of GraphMix(GCN) , we searched over the values in the set $[0.0, 0.1, 0.5, 1.0]$ and found that $0.1$ works best across all the datasets. For $\gamma_{max}$, we searched over the values in the set $[0.1, 1.0, 10.0]$ and found that $0.1$ and $1.0$ works best across all the datasets. Rest of the details for GraphMix(GCN) and GCN are same as Section A.5.1.

### A.5.3 FOR RESULTS REPORTED IN SECTION 4.3

For $\alpha$ of GraphMix(GCN) , we searched over the values in the set $[0.0, 0.1, 0.5, 1.0]$ and found that $0.1$ works best for both the datasets. For $\gamma_{max}$, we searched over the values in the set $[0.1, 1.0, 10.0]$ and found that $0.1$ works best for both the datasets. We conducted all the experiments for 150 epochs. The value of consistency coefficient $\gamma$ (line 13 in Algorithm 1) is increased from 0 to its maximum value $\gamma_{max}$ from epoch 75 to 125 using the sigmoid ramp-up of Mean-Teacher (Tarvainen & Valpola, 2017).

Both for GraphMix(GCN) and GCN, we use Adam optimizer with learning rate $0.01$ and $L2$-decay $0.0$, the number of units in the hidden layer $128$ , dropout rate in the input layer was set to $0.5$.

## A.6 HYPERPARAMETER SELECTION

For each configuration of hyperparameters, we run the experiments with 100 random seeds. We select the hyperparameter configuration which has the best validation accuracy averaged over these 100 trials. With this best hyperparameter configuration, for 100 random seeds, we train the model again and use the validataion set for model selection ( i.e. we report the test accuracy at the epoch which has best validation accuracy.)

## A.7 SOFT-RANK

Let $\mathbf{H}$ be a matrix containing the hidden states of all the samples from a particular class. The Soft-Rank of matrix $\mathbf{H}$ is defined by the sum of the singular values of the matrix divided by the largest singular value. A lower Soft-Rank implies fewer dimensions with substantial variability and it provides a continuous analogue to the notion of rank from matrix algebra. This provides evidence that the concentration of class-specific states observed when using GraphMix in Figure 3 can be measured directly from the hidden states and is not an artifact of the T-SNE visualization.

## A.8 FEATURE VISUALIZATION

We present the 2D visualization of the hidden states learned using GCN and GraphMix(GCN) for Cora, Pubmed and Citeseer datasets in Figure 3. We observe that for Cora and Citeseer, GraphMix learns substantially better hidden states than GCN. For Pubmed, we observe that although there is no clear separation between classes, "Green" and "Red" classes overlap less using the GraphMix, resulting in better hidden states.

### A.8.1 HYPERPARAMETER DETAILS FOR RESULTS IN TABLE 7

For all the experiments we use the standard architecture mentioned in Section A.5 and used Adam optimizer with learning rate 0.001 and 64 hidden units in the hidden layer. For Coauthor-CS and Coauthor-Physics, we trained the network for 2000 epochs. For Cora-Full, we trained the network for 5000 epochs because we observed the training loss of Cora-Full dataset takes longer to converge.

For Coauthor-CS and Coauthor-Physics: We set the input layer dropout rate to 0.5 and weight-decay to 0.0005, both for GCN and GraphMix(GCN) . We did not conduct any hyperparameter search over the GraphMix hyperparameters $\alpha$, $\lambda_{max}$, temparature $T$ and number of random permutations $K$ applied to the input data for GraphMix(GCN) for these two datasets, and set these values to 1.0, 1.0, 0.1 and 10 respectively.

For Cora-Full dataset: We found input layer dropout rate 0.2 and weight-decay 0.0 to be the best for both GCN and GraphMix(GCN) . For GraphMix(GCN) we fixed $\alpha$, temparature $T$ and

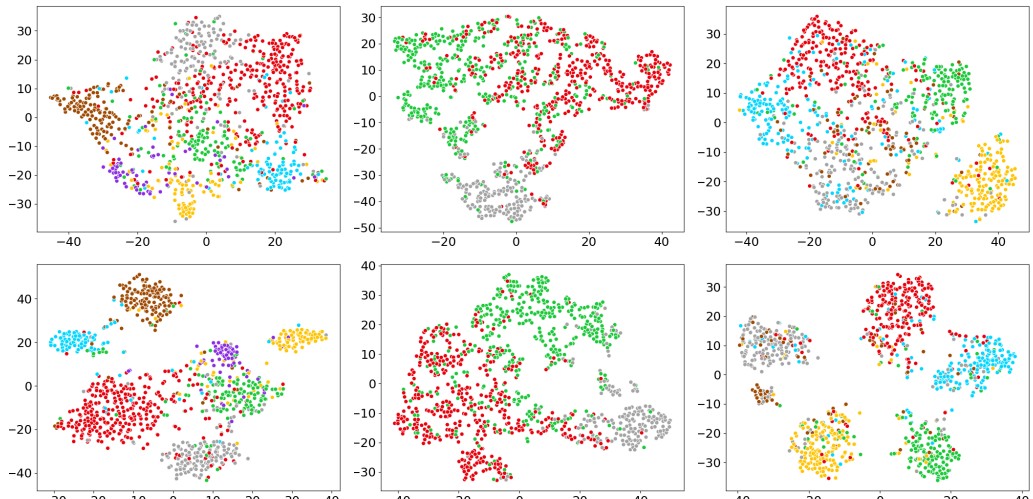

Figure 3: T-SNE of hidden states for Cora (left), Pubmed (middle), and Citeseer (right). Top row is GCN baseline, bottom row is GraphMix.

number of random permutations $K$ to 1.0 0.1 and 10 respectively. For $\lambda_{max}$, we did search over $\{1.0, 10.0, 20.0\}$ and found that 10.0 works best.

For all the GraphMix(GCN) experiments, the value of consistency coefficient $\gamma$ (line 13 in Algorithm 1) is increased from 0 to its maximum value $\gamma_{max}$ from epoch 500 to 1000 using the sigmoid ramp-up of Mean-Teacher (Tarvainen & Valpola, 2017).

