# OpenReview forum: "GraphMix: Regularized Training of Graph Neural Networks for Semi-Supervised Learning"
_ICLR.cc/2020/Conference — Reject_

### Official Review · AnonReviewer3 · 2019-10-23
**Official Blind Review #3**

**Rating:** 3

**Review:**

In this paper, the authors developed GNN by integrating an interpolation based regularization. In particular, the proposed GraphMix model has two components, one GNN and one MLP. For the MLP, the manifold mixup loss is used for training. The two components share parameters so that the GCN training can be regularized by the manifold mixup loss. Moreover, the predicted samples of GCN are used to augment the data of MLP for training in a self-supervised manner. In the experiments, GraphMix was compared with several recent GNN models on several benchmark datasets and demonstrates effectiveness for node classification and link classification.

There are several concerns on this paper.
1. The motivation to integrate interpolation based regularization for GNN is not intuitively clear. The authors may want to further clarify on how can Eq. (2) help improve the classification power of GNN in an intuitive manner.
2. Some algorithm design choices, such as sharpening for entropy minimization, are not well justified. It is not clear why sharpening is selected over psudolabels.
3. The overall technical contribution is somewhat incremental based on previous work on GNN and manifold mixup based regularizations. The model design is heuristic, and lack theoretic analysis, for example, on the parameter sharing based regularization over other choices of regularizations, if any.
4. From the experimental results, the proposed model improves GCN and GAT, but the improvements on other baseline methods are a little bit subtle. The authors may want to further evaluate their framework on those GNNs such as GMNN and Graph U-Net to make the experiments more comprehensive, or justify on why they cannot be implemented using the proposed framework.
5. In Table 2, 3, and 4, it is not clear on why GAT and GraphMix (GAT) are missing. This inconsistency with regard to Table 1 should be justified.
6. In the ablation analysis, if the reason for the suboptimal performance when using EMA is the lack of hyperparameter search, then it is suggested to perform comprehensive hyperparameter search so that the results are more solid. Otherwise, the design choice cannot be well justified.


**Experience Assessment:**

I have published in this field for several years.

**Review Assessment: Checking Correctness Of Derivations And Theory:**

N/A

**Review Assessment: Checking Correctness Of Experiments:**

I carefully checked the experiments.

**Review Assessment: Thoroughness In Paper Reading:**

I read the paper thoroughly.

---

> ### Author Response · Authors · 2019-11-13
> **Response (part 1)**
>
> Thankyou for your valuable comments and feedback. In the following, we address the concerns on point by point basic :
>
> “1. The motivation to integrate interpolation based regularization for GNN is not intuitively clear. The authors may want to further clarify on how can Eq. (2) help improve the classification power of GNN in an intuitive manner. “
>
> Thanks for pointing this out. We have made it more clear in the revised version. The main motivation of this work is to propose a data-augmentation technique (such as Mixup or Manifold Mixup) for Graph Structured data. A naive way to apply such data-augmentation would be to mix the nodes features and their corresponding labels to synthesize new nodes. However, how to connect these synthesized nodes to other nodes of the graph, such that they confirm to the structure of the graph, remains unclear. To alleviate this issue, we proposed to train an auxiliary Fully-connected-net (FCN)     using Manifold Mixup. Note that the FCN only uses the node features ( not the graph structure), thus the Manifold mixup loss in Eq. (2) can be directly used for training the FCN. The Manifold Mixup training of FCN facilitates learning more discriminative node features. How to transfer  these discriminative node features learned by the FCN to the GCN? We apply parameter sharing between FCN and GCN to do so. Using these more discriminative features of nodes and the graph structure, GCN loss is computed in the usual way. Further, since the parameters of GCN and FCN are shared, in the next learning update for training the FCN, the FCN receives more refined representations learned by GCN. In this way both FCN and GCN improve each-others learning akin to Co-training framework. This is discussed in 3.2.3 ( Connection to Co-training).
>
>
> 2. Some algorithm design choices, such as sharpening for entropy minimization, are not well justified. It is not clear why sharpening is selected over psudolabels.
>
> Pseudolabeling technique constructs hard(1-hot) labels for the unlabeled data which has “high-confidence predictions”. Since many of  the unlabeled data samples may have “low-confidence predictions”, they can not be used  in the Pseudolabeling technique. On the other hand, Sharpening [1] does not require “high-confidence predictions” , and thus it can be used for all the unlabelled samples. Thanks for pointing this out, we have made more clear in the revised version of the paper why sharpening is selected over the pseudolabels.
>
> [1]David Berthelot, Nicholas Carlini, Ian Goodfellow, Nicolas Papernot, Avital Oliver, and Colin Raffel. MixMatch: A Holistic Approach to Semi-Supervised Learning.
> arXiv e-prints, art. arXiv:1905.02249, May 2019
>
>
> 3. The overall technical contribution is somewhat incremental based on previous work on GNN and manifold mixup based regularizations. The model design is heuristic, and lack theoretic analysis, for example, on the parameter sharing based regularization over other choices of regularizations, if any.
>
> Much of the recent work in Graph Neural Networks has been focused on designing complex architectures, with little attention to regularization techniques. The main contribution of this paper is to show that the state-of-the-art performance for the semi-supervised node classification can be achieved by using even simple regularization techniques proposed in this paper. This paves the way for attracting more attention to designing better regularization techniques which are tailor-made for the Graph Neural Networks. Furthermore, due to its simplicity, no-additional computation cost and state-of-the-art performance, this method can serve as a strong baseline for future research into designing novel architectures for the Graph Neural Networks.
>
>
> 4. From the experimental results, the proposed model improves GCN and GAT, but the improvements on other baseline methods are a little bit subtle. The authors may want to further evaluate their framework on those GNNs such as GMNN and Graph U-Net to make the experiments more comprehensive, or justify on why they cannot be implemented using the proposed framework.
>
> We evaluated GraphMix framework on Graph-U-Net. The results are included in Table 1 ,which  demonstrate that GraphMix can improve the accuracy of Graph-U-Net (Note that we were not able to reproduce the numbers reported in the Graph-U-Net paper, nevertheless GraphMix improved over the vanilla Graph-U-Net across all the datasets) .
>
> Applying GraphMix to GMNN is not straight forward. The reason is that GMNN is indeed a training framework for existing GNN models instead of being a GNN architecture (e.g., GCN, GAT, Graph U-Net). In this sense, GMNN is similar to GraphMix, which are used for enhancing existing GNN architectures, and it is nontrivial to apply GraphMix to GMNN.

---

> > ### Author Response · Authors · 2019-11-13
> > **Response (Part 2)**
> >
> > 5. In Table 2, 3, and 4, it is not clear on why GAT and GraphMix (GAT) are missing. This inconsistency with regard to Table 1 should be justified.
> >
> > The reason behind having GAT and GraphMix(GAT) in Table1 was to show that GraphMix can improve over other standard architecture (other than GCN such as GAT) as well. Table 2, 3 and investigate other aspects such as “less labeled data” and “random split”  rather than trying to compare against other standard architectures. For completeness, we have added GAT and GraphMix (GAT) results in Table2 and 3 (Table 1 and Table 6 in revised version).
> >
> > For the datasets of Table 4 (Table 2 in the revised version), we were unable to perform GAT and GraphMix(GAT) using a 32GB GPU RAM, due the prohibitive memory requirements of GAT. GAT has O(V^2) memory complexity, V being the number of nodes in the graph.
> >
> >  6. In the ablation analysis, if the reason for the suboptimal performance when using EMA is the lack of hyperparameter search, then it is suggested to perform comprehensive hyperparameter search so that the results are more solid. Otherwise, the design choice cannot be well justified.
> >
> > We have run the extensive hyperparameter search now and we observe that EMA can indeed achieve better results (Table 3).

---

### Official Review · AnonReviewer2 · 2019-10-28
**Official Blind Review #2**

**Rating:** 3

**Review:**

This paper introduces a regularization method for graph neural networks that exploit unlabeled examples. The authors claim that the proposed method for semi-supervised graph learning is based on data augmentation which is efficient and can be applied to different graph neural architectures, that is an architecture-agnostic regularization technique is considered in the paper.

Strength
•	Simple yet effective regularization technique to neural networks for graph-structured data

Weaknesses
•	Weak technical contribution to the problem
•	The performance gain is from mostly the use of a semi-supervised learning approach based on entropy minimization, which has been developed without consideration of graph-structured data.


The paper does not compare the computational complexity of the proposed approach to the baseline approaches. Although it seems the additional computation cost for the proposed approach compared to the baseline such as GCN does not increase dramatically, it would be better if the authors could provide its computational cost as well as performance given the number of perturbed samples.

According to Table 1, self-learning approaches, i.e., GCN (with predicted targets), works comparable to GraphMix. I do not find the benefit of using such a more complicated approach other than simple approaches such as self-learning in the experiment section.

Manifold Mixup is not explained properly. For example, the authors may need to explain two random samples, i.e., (x, y) and (x’, y’) are drawn from the data distribution D, and how it is used in GraphMix more clearly.

Adjacency matrix A is used without its definition.

**Experience Assessment:**

I have read many papers in this area.

**Review Assessment: Checking Correctness Of Derivations And Theory:**

I assessed the sensibility of the derivations and theory.

**Review Assessment: Checking Correctness Of Experiments:**

I carefully checked the experiments.

**Review Assessment: Thoroughness In Paper Reading:**

I read the paper at least twice and used my best judgement in assessing the paper.

---

> ### Author Response · Authors · 2019-11-13
> **Response**
>
> Thanks for your valuable comments and feedback. We have made made the necessary change suggested by you. Following is the more detailed response on point by point basis:
>
>
> "Weak technical contribution to the problem"
>
> (We have addressed this concern in the "common response to all the reviewers" comment above, so you can skip the response to this point if you have already read  "common response to all the reviewers"comment.)
> We agree that the proposed framework in not entirely novel, rather it is a re-synthesis/adaptation/modification of various existing methods. However, it has major advantages such as 1) it can be applied to any underlying GNN that uses a parametric transformation of node feature vectors 2) our experiments show that it improves the test accuracy over a number of underlying GNNs and datasets 3) It has almost no additional computation cost 4) The framework can be implemented on top of an existing GNN using only a few lines of codes. 5) The framework is not very sensitive to the hyperparameters (it improves the test accuracy over the underlying GNNs with even the minimal hyperparameter search)
>
> "The performance gain is from mostly the use of a semi-supervised learning approach based on entropy minimization, which has been developed without consideration of graph-structured data."
>
> Shchur et .al[1] have demonstrated that the performance of various GNN architectures using the  standard Train/validation/Test split of benchmark datasets Cora/Citeseer/Pubmed is significantly different than the random split of the datasets. Based on their observations, to avoid any bais introduced by the standard(fixed) split, they recommend using random splits. Along these lines we conduct our experiments using 10 random Train/Validation/Test splits of the data. Our results in Table1 demonstrate that GraphMix always performs better than existing Semi-supervised learning based approach (GCN(Predicted-targets)), by a sizeable margin.
>
>
> Furthermore, the success of  existing Semi-supervised learning approach such as GCN(predicted-targets) depends heavily on the quality of the  predicted-targets. The quality of the predicted-targets degrades if the number of labeled samples is lesser. To demonstrate this, we conducted a new ablation study, where we limited the number of per-class labels to 10. Results are in Table 3. Our results suggest that the Manifold Mixup training of FCN via shared parameters plays an important role in the success of the proposed framework.
>
> [1]Oleksandr Shchur, Maximilian Mumme, Aleksandar Bojchevski, and Stephan Günnemann. Pitfalls of graph neural network evaluation. CoRR
> , abs/1811.05868, 2018. URL http://arxiv.org/abs/1811.0586
>
>
> “The paper does not compare the computational complexity of the proposed approach to the baseline approaches. Although it seems the additional computation cost for the proposed approach compared to the baseline such as GCN does not increase dramatically, it would be better if the authors could provide its computational cost as well as performance given the number of perturbed samples. “
>
> The proposed framework does not add any significant additional computation cost over the underlying GNN, because the underlying GNN is trained in the standard way and the FCN training requires trivial additional computation cost for computing the predicted-targets (Section 3.2.1 and Section 3.2.2) and the interpolation function ( Mix(a,b) in Section 2.3). We have added this explanation in the revised paper.
>
>
> “According to Table 1, self-learning approaches, i.e., GCN (with predicted targets), works comparable to GraphMix. I do not find the benefit of using such a more complicated approach other than simple approaches such as self-learning in the experiment section.”
>
> According to Table 1 ( Table 5 in the revised version) , GCN (with predicted targets) works better than GraphMix only on Pubmed using the standard split. On Cora and Citeseer, GraphMix is significantly better than GCN (with predicted targets) using the standard split.  As mentioned above, a more justifiable experimental setting is random split of data in train/validation/test sets. In these experiments ( Table 1), GraphMix(GCN) improves over GCN (with predicted targets) across all the datasets, by a significant margin.
>
>
>
>
> “Manifold Mixup is not explained properly. For example, the authors may need to explain two random samples, i.e., (x, y) and (x’, y’) are drawn from the data distribution D, and how it is used in GraphMix more clearly.”
> “Adjacency matrix A is used without its definition.”
>
> We have corrected these mistakes. Thanks for pointing this out.

---

### Official Review · AnonReviewer4 · 2019-11-04
**Official Blind Review #4**

**Rating:** 6

**Review:**

Summary:
This paper presents a data augmentation procedure for semi-supervised learning on graph structured data. In general it is not clear how augmented observations can be incorporated into an existing graph while still preserving the graph structure. Instead of trying to incorporate an augmented dataset into the graph, this paper uses a seperate fully-connected network (FCN) that shares weights with a graph neural net (GNN). The final method also incorporates various components from the literature, such as Mixup and Sharpening. While the literature appears to have been adequately cited, the position of GraphMix relative to previous work could be better dilineated. In my opinion, the main contribution of this paper is the suggestion of using weight-sharing to sidestep the issue of incorporating augmented datapoints into graph-structured data. I am not sufficiently versed in the literature to assess whether the idea is sufficiently novel to warrant acceptance. The idea is simple and appears to improve the performance of Graph Convolutional Nets (GCNs), so I am leaning towards acceptance.

Rating Justification:
Efficient data-augmentation procedures are an important area of research. The relative simplicity and generality of GraphMix is appealing. I would give the paper a higher score if the authors showed that GraphMix(Graph U-net) was an improvement over Graph U-net, or if it was made more clear that some substantial benefit is derived from using Mixup features.

Additional Comments:
1. Based on the ablation study it seems that Mixup actually plays a very minor role in the overall success of the procedure. I would be curious to see the t-SNE visualization of the GraphMix(GCN) \ Mixup features in order to determine how much of the cluster separation is due to Mixup specifically. I understand that previous work has suggested that Mixup features are superior for discrimination than normal features, but in this work specifically the evidence for this assertion is fairly weak.

2. A clearer distinction between GraphMix(GCN) and GCN (with predicted targets) would be very helpful, especially since GCN (with predicted targets) actually performs the best on the standard PubMed splits. Why were the results for GCN (with predicted targets) not included in Table 2?

**Experience Assessment:**

I do not know much about this area.

**Review Assessment: Checking Correctness Of Derivations And Theory:**

I assessed the sensibility of the derivations and theory.

**Review Assessment: Checking Correctness Of Experiments:**

I assessed the sensibility of the experiments.

**Review Assessment: Thoroughness In Paper Reading:**

I read the paper at least twice and used my best judgement in assessing the paper.

---

> ### Author Response · Authors · 2019-11-13
> **Response**
>
> Thankyou for your valuable comments and feedback. In the common response to all the reviewers above,  we have listed the main advantages of this approach and the reasons we believe  this can be a useful work for the research community. In the following we address the concerns point by point:
>
> “Rating Justification: Efficient data-augmentation procedures are an important area of research. The relative simplicity and generality of GraphMix is appealing. I would give the paper a higher score if the authors showed that GraphMix(Graph U-net) was an improvement over Graph U-net, or if it was made more clear that some substantial benefit is derived from using Mixup features.”
>
> We have added results for Graph-U-Net in Table 1. Our experiments demonstrate the GraphMix(Graph-U-Net) improves the results of the vanilla Graph-U-Net.  For “substantial benefit is derived from using Mixup features”, please see below.
>
>
>
> "1. Based on the ablation study it seems that Mixup actually plays a very minor role in the overall success of the procedure. I would be curious to see the t-SNE visualization of the GraphMix(GCN) \ Mixup features in order to determine how much of the cluster separation is due to Mixup specifically. I understand that previous work has suggested that Mixup features are superior for discrimination than normal features, but in this work specifically the evidence for this assertion is fairly weak."
>
> We have added Figure 2B which has the t-SNE visualization of  GCN (trained with the predicted targets), which is essentially  GraphMix(GCN) \ Mixup features.  Figure 2C has   t-SNE visualization of  GraphMix(GCN). By comparing these two plots, we can observe that the GraphMix(GCN) obtains better cluster separation. This can be also demonstrated with the soft-rank of class-specific features ( Figure 2D) .
>
> Furthermore, we would like to point out that, Mixup plays even more important role when the labeled samples are very small.  To demonstrate this, we conducted a new ablation study, where we limited the number of per-class labels to 10. Results are in Table 3. Our results suggest that the Manifold Mixup training of FCN via shared parameters plays an important role in the success of the proposed framework.
>
>
>  "2. A clearer distinction between GraphMix(GCN) and GCN (with predicted targets) would be very helpful, especially since GCN (with predicted targets) actually performs the best on the standard PubMed splits."
>
> We have improved the writing to  make this clear in the revised version.
>
>
> "3. Why were the results for GCN (with predicted targets) not included in Table 2?"
>
> We have added the results for GCN (with predicted targets) in Table 2 ( Table 1 in the revised version).

---

### Public Comment · ~Bao_Wang1 · 2019-10-19
**A related paper on graph interpolation function**

Hi, it is a very cool idea to apply interpolation-based regularization, and I like it. Here I would like to point out a few related papers.

1. B. Wang, et al. Deep Neural Nets with Interpolating Function as Output Activation, NeurIPS 2018.

2. B. Wang, et al. Adversarial Defense via Data Dependent Activation Function and Total Variation Minimization, arXiv:1809.08516 2018

3. B. Wang, et al. Graph Interpolating Activation Improves Both Natural and Robust Accuracies in Data-Efficient Deep Learning, arXiv:1907.06800 2019.

Thanks for your attention.

---

> ### Author Response · Authors · 2019-10-21
> **Thanks for the references**
>
> Hello,
>
> Thanks for pointing out the references. We will refer to them adequately in the revised version of the paper.

---

> > ### Public Comment · ~Bao_Wang1 · 2019-10-26
> > **Thanks.**
> >
> > We thank the authors' attention in these papers.

---

### Author Response · Authors · 2019-11-13
**Common response for all the reviewers: Summary of the changes made**

We thank the reviewers for their helpful comments and suggestions. All the reviewers agreed that Data-augmentation based regularization for Graph Structured data is an important problem and the proposed framework is simple and effective approach to address this problem, with experimental results using a variety of architectures and datasets demonstrating its effectiveness.

We have incorporated all proposed changes and performed all suggested experiments. We believe that the quality of the paper has been improved and our contribution is more  clear now.

As pointed out by reviwer 2 and 3, we have made it more clear in the revised version of the paper that the proposed framework is a re-synthesis/adaptation/modification of various existing methods from the literature for Graph structured data, rather than being an entirely new method. However, we would like to point out the major advantages of this work:  1) it can be applied to any underlying GNN that uses a parametric transformation of node feature vectors 2) our experiments show that it improves the test accuracy over a number of underlying state-of-the-art GNNs and datasets 3) It has almost no additional computation cost 4) The framework can be implemented on top of an existing GNN using only a few lines of codes. 5) The framework is not very sensitive to the hyperparameters (it improves the test accuracy over the underlying GNNs with even the minimal hyperparameter search).

This serves two purposes: (1) Due to its effectiveness, our method paves the way for attracting more attention to designing better data-augmentation techniques for the Graph Neural Networks. (2) Due to its simplicity, no-additional computation cost and state-of-the-art performance, this method can serve as a strong baseline for future research into designing novel architectures for the Graph Neural Networks.

The summary of the changes made in the revised version is as follows:

1: The writing has been improved to clearly communicate the motivation of the proposed framework.
2. We have added Graph-U-Net experiments.  Our results show that GraphMix can improve the results over vanilla Graph-U-Net. ( Note that despite our best efforts, we were not able to reproduce the results in the Graph-U-Net paper, so our results for vanilla Graph-U-net are not similar to the Graph-U-Net paper).
3. To maintain the consistency in the tables, we have added GCN(predicted-targets), GAT and GraphMix(GAT) results in all the tables where necessary.
4. We have added a t-SNE visualization of GCN(predicted-targets) to demonstrate how much cluster separation happens because of Manifold Mixup training of FCN.
5. We have added explanation for design choices such as why “sharpening” is used instead of “Psuedo-labels”.


We address the concerns of each reviewer individually in the following comments.

---

### Decision · Program_Chairs · 2019-12-19

**Decision:**

Reject

**Comment:**

The authors integrate an interpolation based regularization to develop a graph neural network for semi-supervised learning. While reviewers enjoyed the paper, and the authors have provided a thoughtful response, there were remaining questions about clarity of presentation and novelty remaining after the rebuttal period. The authors are encouraged to continue with this work, accounting for reviewer comments in future revisions.